# Animal life in the shallow subseafloor crust at deep-sea hydrothermal vents

Monika Bright [1,12] ✉, Sabine Gollner [2,12] ✉, André Luiz de Oliveira [3], Salvador Espada-Hinojosa [1], Avery Fulford[4,5], Ian Vincent Hughes [6], Stephane Hourdez[7], Clarissa Karthäuser [4], Ingrid Kolar[1], Nicole Krause[1], Victor Le Layec[7], Tihomir Makovec[8], Alessandro Messora[2], Jessica Mitchell [6], Philipp Pröts [1], Ivonne Rodríguez-Ramírez [9], Fanny Sieler[2], Stefan M. Sievert [4], Jan Steger [10], Tinkara Tinta [8], Teresa Rosa Maria Winter [1], Zach Bright[11], Russel Coffield[11], Carl Hill[11], Kris Ingram[11] & Alex Paris[11]

It was once believed that only microbes and viruses inhabited the subseafloor crust beneath hydrothermal vents. Yet, on the seafloor, animals like the giant tubeworm *Riftia pachyptila* thrive. Their larvae are thought to disperse in the water column, despite never being observed there. We hypothesized that these larvae travel through the subseafloor via vent fluids. In our exploration, lifting lobate lava shelves revealed adult tubeworms and other vent animals in subseafloor cavities. The discovery of vent endemic animals below the visible seafloor shows that the seafloor and subseafloor faunal communities are connected. The presence of adult tubeworms suggests larval dispersal through the recharge zone of the hydrothermal circulation system. Given that many of these animals are host to dense bacterial communities that oxidize reduced chemicals and fix carbon, the extension of animal habitats into the subseafloor has implications for local and regional geochemical flux measurements. These findings underscore the need for protecting vents, as the extent of these habitats has yet to be fully ascertained.

The East Pacific Rise (EPR) is a volcanically active, fast-spreading ridge with numerous hydrothermal vent fields[1]. The last two eruptions of the EPR segment near 9°50′N[2] in 1991–1992[3] and 2005–2006[4] revealed a sequential megafauna colonization pattern at new vent fields from siboglinid tubeworms to bathymodiolin mussels[5–8] thriving at distinct vent fluid regimes, and providing habitat for low diverse but biomass-rich animal communities[6,9,10]. Rapid colonization by these animals

suggests efficient larval dispersal, with larvae assumed to be transported through bottom, ridge and ocean currents[5,6,10–14] before they settle at vents through downwards swimming or sinking[8,12,13].

The sessile siboglinid *Riftia pachyptila* (Polychaeta, Siboglinidae) and its two smaller relatives *Tevnia jerichonana* and *Oasisia alvinae* thrive at the vigorous diffuse-flow vents of the EPR[9]. As adults, the three tubeworm species lose their mouth and gut, relying completely

[1]University of Vienna, Department of Functional and Evolutionary Ecology, Vienna, Austria. [2]Royal Netherlands Institute for Sea Research (NIOZ), 't Horntje, The Netherlands. [3]Max-Planck Institute for Marine Microbiology, Bremen, Germany. [4]Biology Department, Woods Hole Oceanographic Institution, Woods Hole, USA. [5]MIT-WHOI Joint Program in Oceanography/Applied Ocean Science & Engineering, Cambridge and Woods Hole, USA. [6]Harvard University, Department of Organismic and Evolutionary Biology, Cambridge, USA. [7]UMR 8222 LECOB, CNRS-Sorbonne Université, Observatoire Océanologique de Banyuls, Banyuls-sur-Mer, France. [8]Marine Biology Station Piran, National Institute of Biology, Piran, Slovenia. [9]University of Costa Rica, School of Biology, San Pedro, Costa Rica. [10]University of Vienna, Department of Palaeontology, Vienna, Austria. [11]Schmidt Ocean Institute, Palo Alto, USA. [12]These authors contributed equally: Monika Bright, Sabine Gollner. ✉e-mail: monika.bright@univie.ac.at; sabine.gollner@nioz.nl

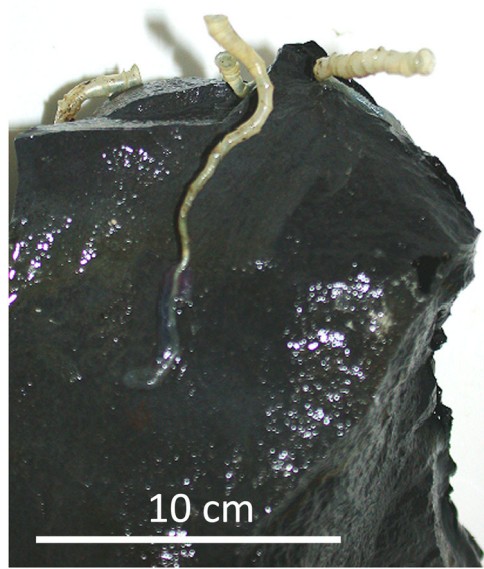

**Fig. 1 | *Oasisia alvinae* colonizing cracks in basalt.** Several specimens found to grow in crack of basalt upwards from ~10 cm beneath the seafloor surface; sampled at Tica, 9.839979° N, 104.2916175° W, ~2515 m depth, Alvin Dive 3849, December 7, 2002. cm = centimeter.

on their shared single gammaproteobacterial symbiont, *Candidatus* Endoriftia persephone, for nutrition[15,16]. Fertilized symbiont-free eggs are released into the environment and have been detected together with embryos in close proximity to tubeworm clumps[17]. Experiments, conducted both in situ and in pressure vessels, revealed that embryos develop into trochophore larvae filled with lipid storage products to survive in the water column for about one month[18]. Embryos are cold stenotherms and barophilic, which suggests that they may disperse over long distances in the deep sea[19]. Upon settlement at vents[20], free-living symbionts[21] colonize the larvae through infecting their skin[20], while they escape back into the environment when their host dies[22].

Tubeworm larval retention within a vent field was suggested[23], but from where and how these larvae arrive to settle at a vent is unknown, as they have never been detected in the water column[12,24]. Although arrival in bottom water, as posited for other animals, seems likely, since larvae cannot swim against the currents but are moved upwards with the vent flow[10], we propose that larvae may also get entrained with seawater into the ocean crust[25], where they transit through the subseafloor to finally settle at vents, arriving from beneath within the vent fluid discharged through the cracks to the surface. Transit of larvae through the shallow Earth's crust in porous volcanic rocks, channels and small subsurface cavities has been suggested (unpublished illustration commissioned by Van Dover 1997,[26]), but has to our knowledge never been investigated. Over the past 15 years, adult *O. alvinae* and unidentified tubeworm juveniles were regularly encountered below *R. pachyptila* clumps in tiny cracks of basalt down to 10 cm beneath the seafloor (Fig. 1), although it seems unlikely that the minute tubeworm larvae can actively swim or crawl 10 cm against the vent flow discharged at the seafloor, even when flow is variable and may be low. Further, in contrast to other tubeworms (e.g. *Lamellibrachia barhami*[27]), *O. alvinae* and *R. pachyptila* are not known to develop roots to extend their body into the subseafloor[9].

To test the hypothesis of larval transit through the shallow Earth's crust we chose to drill small holes into the exposed igneous rock and lift some lava shelves. This rocky subseafloor at ridge axes is characterized by hydrothermally driven fluid circulation and predicted to restrict the subseafloor biosphere to several meters below the seafloor[28–30]. Because it has been technically impossible to drill deep holes into the crust[28], subseafloor microbes and viruses have been

sampled from hydrothermal fluid samples emerging the seafloor[28,31–33]. While the subseafloor microbial and viral biosphere at deep-sea vents has been described, we show that animal life also exists in this shallow rocky subseafloor province. These findings support our hypothesis that there is larval dispersal within the crustal subseafloor, but also expands the known macrofaunal biosphere.

## Results and discussion
### Subseafloor cavities habitat
We chose a vent site, which we named Fava Flow Suburbs, at the 9°50'N EPR at 2515 m water depth. We selected small, discrete clumps of *R. pachyptila* and *O. alvinae* tubeworms ($N = 1$), a mix of tubeworms and *Bathymodiolus thermophilus* mussels ($N = 3$), or mussels ($N = 2$) along a hydrothermal flux gradient[28] growing above cracks on lobate lava (Figs. 2 and 3 and Table 1). While attempting an in situ experiment to demonstrate larval settlement from beneath the surface, we discovered animal life in hydrothermal fluid-filled cavities beneath the surface of the crust (Supplementary Movie 1).

Subseafloor cavities between several shelves of lava plates are common in near-vent lobate lavas[28,34–36]. They are suggested to develop between the upper and the lower crust of lava flow due to condensation of trapped vapor of seawater at fast and intermediate spreading centers[34–36]. At Fava Flow Suburbs, we used the ROV SuBastian to lift whole upper lobate lava shelves and discovered caves of ~10 cm height below the ~10–15 cm thick lava shelf. In situ observations at 9°50'N EPR further showed that several shelves of lava plates on top of each other can separate fluid-filled cavities in the subseafloor[34] (Fig. 2b). These geological phenomena are common at fast and intermediate spreading centers[34], e.g. ~10 cm thick upper shelves and up to ~40 cm high cavities were observed at the Juan de Fuca Ridge[34]. Subseafloor cavity systems are suggested to build a reservoir of warm fluids, which escape through cracks in the basalt, nourishing vents at the seafloor[37,38]. Subsurface microbes have been described from fluid samples of deep-sea hydrothermal vents[31,32], including the diffuse flow vent Crab Spa[33] in the vicinity of Fava Flow Suburbs. Here we report, to our knowledge for the first time, the discovery of animals excavated from fluid-filled, shallow cavities in the subseafloor of deep-sea hydrothermal vents.

To constrain the subseafloor conditions in the cavities of the lobate lava, we measured the in situ temperature and took samples of the vent fluid through a small hole drilled with a chisel prior to lifting the lava shelf (Fig. 4; Table 1). In total, six caves were studied of which five were ~10 cm high and inhabited by macroscopically visible animals (Fig. 2c). The single visually uninhabited cavity with three mussels at the surface was only 5 cm high. The five inhabited cavities exhibited maximum temperature of $18.1 \pm 7.1$ °C (mean ± standard deviation). Temperatures in the cavities and among tubeworms/mussels on the seafloor surface were similar ($p = 0.2$), but significantly different for vent surface habitat and ambient temperatures ($p = 0.005$), and subsurface habitat and ambient temperatures ($p = 0.004$). The abiotic conditions inside the cavities, including a temperature of $18.1 \pm 7.1$ °C, a pH $6.1 \pm 0.4$, a salinity $33.8 \pm 0.4$, a minimum concentration of $136 \pm 165$ μmolL$^{-1}$ H$_2$S [i.e., sum of all forms of dissolved sulfide[39]], and maximum $63 \pm 25$ μmolL$^{-1}$ oxygen concentrations (Table 1), were all well in the range of the temporally variable conditions we measured at the surface among animal clumps as well as previously found in tubeworm clumps[39–42]. This clearly supports the shallow crustal subseafloor as a suitable habitat for vent animals[26].

### Tubeworms in subseafloor cavities
Sessile siboglinids were found in five cavities. We note that even when mussels were present at the seafloor surface above four of these cavities, they were not visible in the subseafloor cavities. The most abundant tubeworm species, growing from the roof of all five caves and often wrapped around lava drips (Fig. 2d–h), was *O. alvinae* with

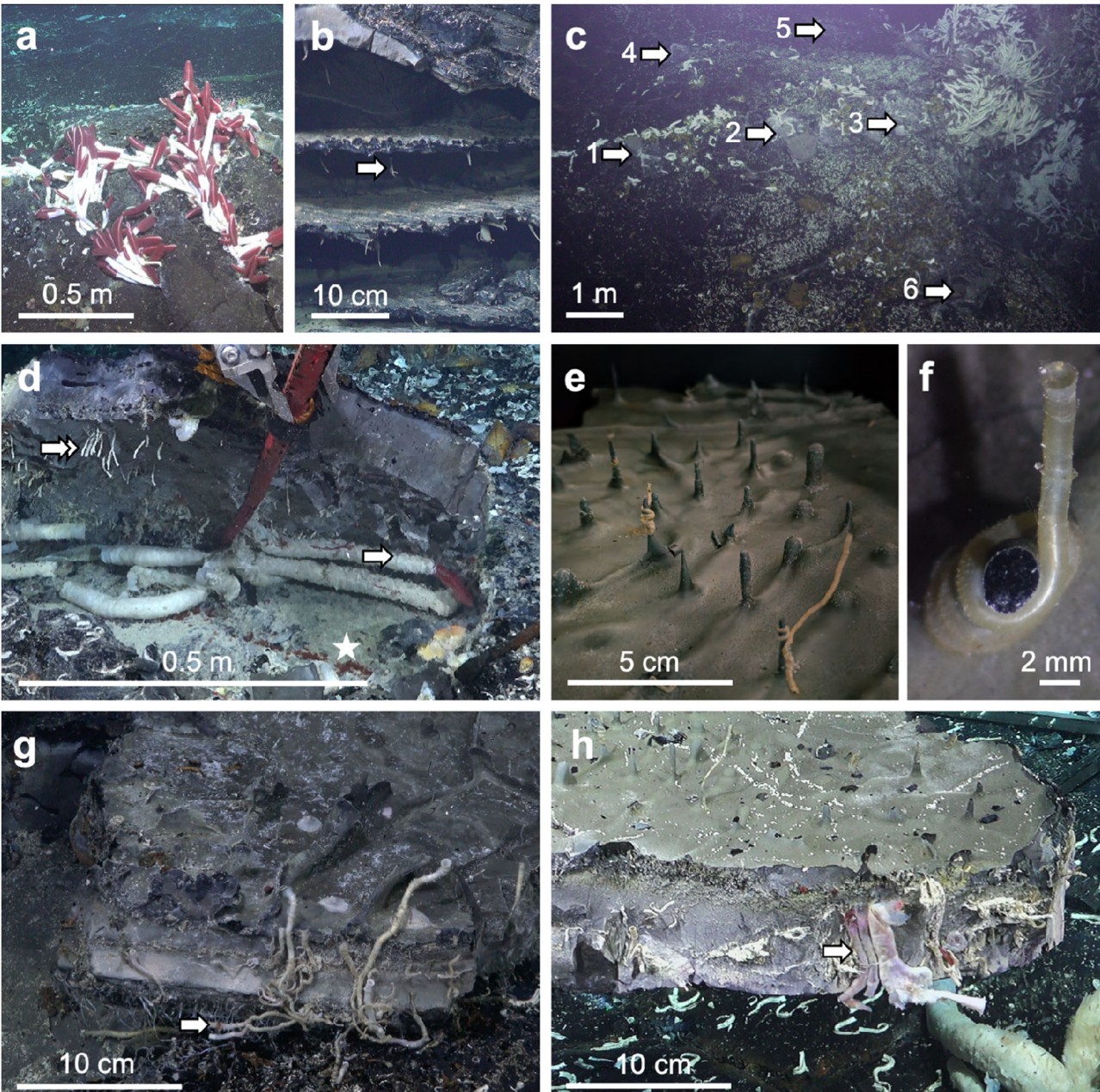

**Fig. 2 | Seafloor surface and crustal subseafloor vents at Fava Flow Suburbs, 9°50′N EPR. a** Tubeworm clumps growing on lobate lava shelf at surface and in cracks (dive S0543). **b** Lobate lava with three uppermost shelves and cavities shown, lava drips (arrow) at the roof of shelves (dive S0560). **c** Six subsurface cavities visible after lifting of lava shelves (recovery dives 1, S0552; 2, S0554; 3, S0556; 4, S0557; 5, S0558; 6, S0560). **d** Opened subseafloor cavity with bottom covered by white microbial mat and live *Paralvinella* spp. (asterisk) and large *Riftia pachyptila* (arrow), roof with alive and tubes of *Oasisia alvinae* (double arrow) (dive S0556). **e,f** Recovered lava shelf shown upside-down, with *O. alvinae* wrapped around lava drips protruding from the ceiling of the opened crustal subseafloor cavity (dive S0552). **g** Upside-down lava shelf with *O. alvinae* mostly growing from crack downwards into cavity (dive S0560). **h** Upside-down lava shelf with *R. pachyptila* growing from crack upwards towards the seafloor (dive S0552). m = meter; cm = centimeter; mm = millimeter.

record sizes of 20 cm tube length, exceeding the ones described from the surface[43] and indicating that these large specimens were adult. In three of these cavities, live specimens were identified alongside empty tubes (Fig. 2e). Living *R. pachyptila* was also found, co-occurring with *O. alvinae* in two cavities, one of them filled with six *R. pachyptila* specimens up to 50 cm tube length (Fig. 2d). Four of the recovered specimens between 30 and 41 cm body length, were clearly adult. Two males had testis with sperm and two females contained large eggs in their gonads, demonstrating that reproduction may happen in the subseafloor cavities. Further, the cracks through which vent fluid leaked, were also inhabited. *O. alvinae* grew either from the shelf

ceiling downwards into the cavity (Fig. 2g) or upwards towards the surface (Fig. 2h). *R. pachyptila* was only growing upwards through the crack, most often extending with the plume above the surface (Fig. 2a, h).

The sampling site Fava Flow Suburbs was characterized by relatively flat surfaces with many visible cracks sparsely colonized by tubeworms and mussels. This allowed us to quantify the animals at the surface and below. While the density (0.16 m⁻²) of live *R. pachyptila* was higher at the surface (5.0 ± 2.8, mean ± standard deviation) and in the cracks (4.0 ± 3.2) than at the subseafloor (1.0 ± 1.5) a different trend was observed for live *O. alvinae* being similar in all three areas (surface

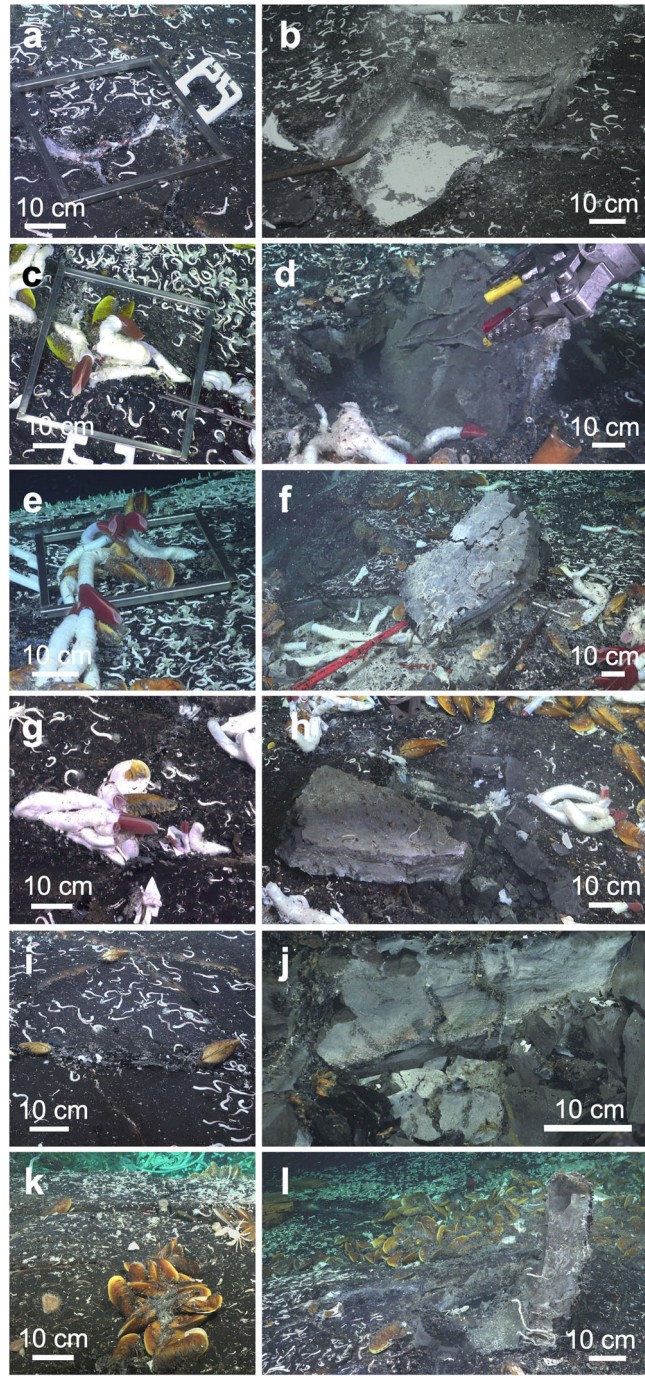

**Fig. 3 | Seafloor surface hydrothermal vents and respective excavation into cavity beneath lobate lava shelf. a** Small *Riftia pachyptila* and *Oasisia alvinae* tubeworms growing in crack of lobate lava (dive S0543) and **b** lifted lava shelf seven days later (dive S0552, recovery dive #1); metal frame 40 × 40 cm. **c** *R. pachyptila* and *O. alvinae* tubeworms and *Bathymodiolus thermophilus* mussels at seafloor surface (dive S0545) and **d** lifted lava shelf seven days later (dive S0554, recovery dive #2). **e** *R. pachyptila* and *O. alvinae* tubeworms and *B. thermophilus* mussels (dive S0547) and **f** lifted lava shelf seven days later (dive S0556, recovery dive #3). **g** *R. pachyptila* and *O. alvinae* tubeworms and *B. thermophilus* mussels and **h** excavation (dive S0557, recovery dive #4). **i** *B. thermophilus* mussels and **j** excavation (dive S0558, recovery dive #5). **k** *B. thermophilus* mussels at seafloor surface and **l** excavation (dive S0560) (recovery dive #6). cm = centimeter.

$3.0 \pm 2.1$, cracks $4.0 \pm 5.5$, subsurface $2.0 \pm 2.7$). The high densities of *O. alvinae* tubes in the cracks ($35.0 \pm 7.1$) and in the subsurface ($61.0 \pm 92.8$), however, were unexpected, indicating a shift from environmental conditions that initially allowed larvae to settle and

grow until unfavorable conditions lead to death. In close vicinity but located in the axial summit trough of the 9°50′N EPR region very high density of *R. pachyptila* up to 2000 per m² were observed during this cruise and were also reported[7,44]. What lives below such lush animal communities remains to be studied.

Our findings indicate that for siboglinids and their obligate endosymbionts, the subseafloor can be a transient as well as a permanent habitat, depending on the settlement location. Larvae can disperse in cavities to potentially colonize lava cracks and the seafloor, or even settle and grow to adults and thereby proliferate as permanent fauna in the shallow crustal subseafloor vents (Fig. 5). Apparently, there is a continuous habitat for tubeworms from diffuse flow subseafloor cavities to the seafloor surface via discharge cracks in the lava shelves. We propose that tubeworm larvae are entrained in the circulation system, which flows from cold bottom seawater intruding into porous volcanic rock (the recharge zone) to the mixing of seawater with hydrothermal vent fluid and the discharge of fluid at the seafloor surface (Fig. 5). Larvae recruit wherever the conditions are suitable along this transit. The discovery of a cavity with about 80 *O. alvinae* specimens on <0.25 m² with similar sizes growing next to and on top of each other (Fig. 3, Table 1, recovery dive S0557) suggests that larvae concentrate underground during this circulation. This process may also facilitate rapid colonization of vents after a volcanic eruption and influence succession[6,7,22].

### Mobile animals in subseafloor cavities
Mobile and semi-sessile vent endemic polychaetes and gastropods[45,46] were also found in four cavities (Fig. 6). Smaller animals and microbes were sampled but detailed analyzes have yet to be carried out. Many individuals of *Paralvinella* spp. (~30 ind. 0.25 m⁻²) were thriving below several large *R. pachyptila* at the bottom of one cave overgrown by a white microbial mat on which they are known to feed[45]. Other deposit feeders detected were the polychaete worm *Nicomache arwidssoni* and limpets, including *Lepetodrilus* spp. and the semi-sessile *Neomphalus fretterae*[45,46]. Further, the carnivorous and scavenging polychaetes *Nereis sandersi* and *Archinome rosacea* were observed[45,46], both occurring in body sizes that are typical for adults[45]. Some animals were also found in the cracks of the basalt (Table 1). These first findings of a subset of vent animals from diffuse-flow vents comprising all trophic levels indicate connectivity with the communities living at the surface. To our current knowledge, all these species exhibit pelagic larval dispersal, indicating that they may have colonized the subseafloor through the recharge zone. In contrast to sessile animals, we confirm that at least some of the mobile animals migrate in and out of the subseafloor by crawling through the cracks in the lobate lava, being able to enter the cavities via the discharge zone (Fig. 5).

### Implications of animal life in subseafloor cavities
The discovery of animal life beneath the surface of the Earth's crust raises questions concerning the extent of these ecosystems, which is larger than what can be seen on the seafloor surface. In contrast to a specialized and isolated subsurface nematode species that has been discovered kilometers deep in terrestrial aquifers[47], we detected a typical surface vent community in subseafloor cavities at marine deep-sea hydrothermal vents. This connectivity of vent subseafloor and seafloor surface habitats may be essential to persist and maintain local and regional biodiversity where the habitat is patchy and transient. The proposed vent subseafloor connectivity model with larval dispersal via the crustal subseafloor adds a new dimension to the known larval dispersal of oceanic, ridge-controlled, and bottom-currents[13]. Further, it extends potential migration routes of mobile vent fauna into the crustal subseafloor.

In addition, given that many of these animals are host to dense bacterial communities that oxidize reduced chemicals and fix carbon, the extension of animal habitats into the subsurface has implications

**Table 1 | Excavations into crustal subseafloor hydrothermal vents at Fava Flow Suburbs at 9°50′N EPR region in 2023**

| ROV recovery dive # | S0552 | S0554 | S0556 | S0557 | S0558 | S0560 |
|---|---|---|---|---|---|---|
| Fig. 2c | #1 | #2 | #3 | #4 | #5 | #6 |
| Fig. 3 | a,b | c,d | e,f | g,h | i,j | k,l |
| Date | 13.07.23 | 15.07.23 | 17.07.23 | 18.07.23 | 19.07.23 | 21.07.23 |
| Longitude | 9.840237 | 9.840121 | 9.840075 | 9.840133 | 9.840238 | 9.840108 |
| Latitude | −104.291860 | −104.291818 | −104.291812 | −104.291874 | −104.292038 | −104.291805 |
| Depth (m) | 2515 | 2515 | 2515 | 2515 | 2512 | 2515 |
| Ambient T (°C) | 1.9 | 1.9 | 1.9 | 2.0 | 2.0 | 2.1 |
| Ambient salinity | 34.7 | 34.7 | 34.7 | 34.7 | 34.7 | 34.7 |
| Ambient oxygen ($\mu mol \cdot L^{-1}$) | 107 | 106 | 105 | 106 | 105 | N/A |
| Surface max. T (°C) | 19.7 | 10.4 | 22.7 | 2.1 | 9.5 | 9.1 |
| Subseafloor max. T (°C) | 24.9 | 24.0 | 24.9 | 11.6 | 7.0 | 10.0 |
| Subs. salinity | 33.8 | 34.0 | 33.3 | 34.0 | 34.4 | 33.8 |
| Subs. pH | 6.4 | 5.7 | 5.5 | 6.5 | 6.5 | 6.2 |
| Subs. min. $H_2S$ ($\mu mol \cdot L^{-1}$)[a] | N/A | 197 | 401 | 14 | 20 | 25 |
| Subs. max. oxygen ($\mu mol \cdot L^{-1}$) | N/A | N/A | 50 | 91 | 115 | 47 |
| ~ Lava shelf thickness (cm) | 10 | 12 | 11 | 15 | 10 | 12 |
| ~ Lava plate surface ($m^2$) | 0.13 | 0.21 | 0.24 | 0.08 | 0.07 | 0.06 |
| ~ Cavity height (cm) | 10 | 10 | 10 | 10 | 5 | 10 |
| **Animals at vent surface[b]** | | | | | | |
| *Oasisia alvinae* | 5 | 4 | 3 | 1 | 0 | 0 |
| *Riftia pachyptila* | 6 | 5 | 7 | 6 | 0 | 0 |
| *Bathymodiolus thermophilus* | 0 | 3 | 5 | 2 | 4 | 22 |
| **Animals in lava crack[b]** | | | | | | |
| *Oasisia alvinae* | 0 | N/A[d] | N/A[d] | 1 | 0 | 10 |
| *Oasisia* tubes | 30 | N/A[d] | N/A[d] | N/A[d] | 0 | 40 |
| *Riftia pachyptila* | 6 | 5 | N/A[d] | N/A[d] | 0 | 0 |
| *Lepetodrilus* spp. | 2 | 1 | 0 | 0 | 0 | 0 |
| *Nereis sandersi* | 1 | 0 | 0 | 1 | 0 | 1 |
| *Paralvinella* spp. | 0 | 0 | 1 | 0 | 0 | 0 |
| **Animals in subseafloor cavities[c]** | | | | | | |
| *Oasisia alvinae* | 0 | 0.8 | 0.7 | 2 | 0 | 5.3 |
| *Oasisia* tubes | 12.3 | 15.2 | 53.3 | 200 | 0 | 16 |
| *Riftia pachyptila* | 0 | 0 | 3.3 | 2 | 0 | 0 |
| *Archinome rosacaea* | 0 | 0.8 | 0 | 0 | 0 | 0 |
| *Branchinotogluma* spp. | 0 | 0.8 | 0.7 | 0 | 0 | 2.7 |
| *Lepetodrilus* spp. | 0 | 1.5 | 0.7 | 0 | 0 | 0 |
| *Lepidonotopodium williamsae* | 0 | 0 | 0.7 | 0 | 0 | 2.7 |
| *Neomphalus fretterae* | 0 | 0 | 0 | 0 | 0 | 10.7 |
| *Nereis sandersi* | 0 | 2.3 | 0 | 0 | 0 | 2.7 |
| *Nicomache arwidssoni* | 1.2 | 0 | 0 | 0 | 0 | 5.3 |
| *Paralvinella* spp. | 0 | 0 | 33.3 | 0 | 0 | 0 |

Abiotic and biotic characteristics of subseafloor (subs.) and surface habitats.

[a] sum of all forms of dissolved sulfide; short sulfide.

[b] custom-made aluminum frame (0.16 m²) placed on the surface over a crack in the lava shelves, animals on the surface and in the crack were counted.

[c] total numbers of counted animals in subseafloor cavities beneath lava shelves estimated to 0.16 m⁻².

[d] present but not counted.

for local and regional geochemical flux measurements[48–50]. Whilst the three-dimensional extent of cavities filled with low temperature fluids is not quantified, the geological phenomena of hollow lobate lava are common[35], and the vent crustal subseafloor habitat may range from the uppermost cavity through several lava shelves down to the lower floors of the subsurface[34]. Due to increase in temperature, it has been predicted that life should be restricted to several meters below the seafloor[28–30]. The current hydrothermal circulation model at the 9°50′N EPR region estimates 90% of hydrothermal vent fluids to be discharged as low temperature fluids at the seafloor[51].

The study of the subseafloor biosphere for animal life has just begun. These efforts will lead to a better understanding of hydrothermal vent biogeochemistry, ecology, and evolution and its impact on global biodiversity, and connectivity, potentially leading to better management of seafloor surface and crustal subseafloor hydrothermal vents. The uniqueness of active hydrothermal vents is well recognized, and protection against potential future anthropogenic impact such as deep-sea mining has been suggested or is in place[52–54]. The discovery of animal habitats in the crustal subseafloor, the extent of which is currently unknown, increases the urgency of such protections.

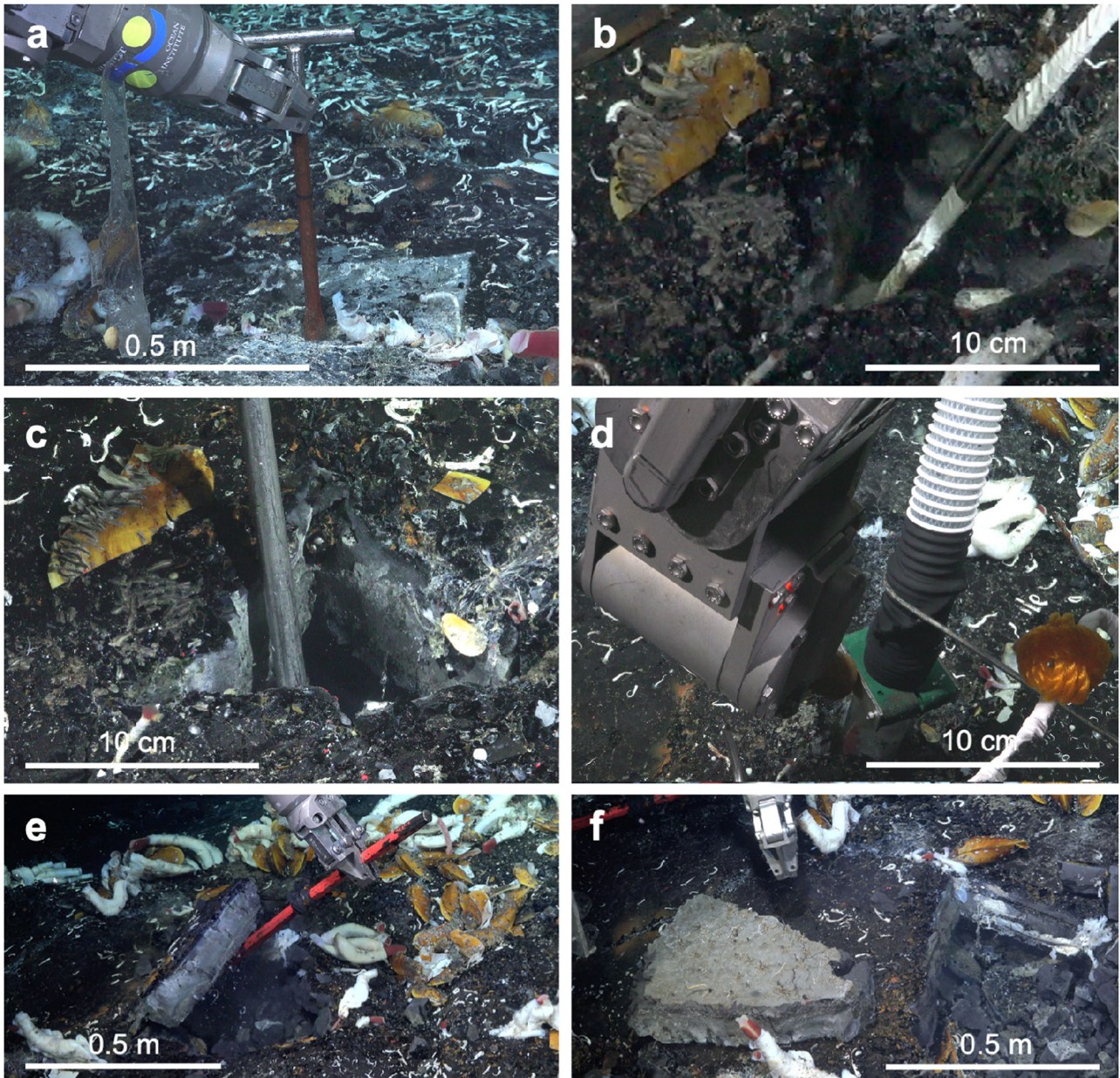

**Fig. 4 | Sampling and excavation scheme to open crustal subseafloor cavity, dive S0557. a** Drilling of a small hole of ~2 cm diameter with a chisel to widen an existing crack in the lava crust. **b** Measurement of in situ temperature in about 20 cm depth below the seafloor surface and sampling of 75 L fluid from the subseafloor cavity. **c** Widening the hole to ~10 cm in diameter. **d** Suction sampling the crustal subseafloor community with 32 μm fine mesh. **e** Lifting of lava shelf. **f** Flipping of lava shelf. m = meter; cm = centimeter.

## Methods

We confirm that our research complies with all relevant regulations for obtaining biological material, collected in areas beyond national jurisdiction, and imported to and exported from Panama and sent to Austria and the Netherlands (Import and export permit to and from Panama, Ministerio de Desarrollo agropecuario (215390 – 215391, 215392–215398), and Ministerio de Ambiente (PA-05-ARB-114-2023; PA-05-ARB-131-2023); import permit from Panama to Austria, Federal Ministry of Social Affairs, Health, Care and Consumer Protection (2023-0.250.333); import permit from Panama to Netherlands, NVWA (NVWA-0166861)).

### Collection of subseafloor fluids and lobate lava

ROV *SuBastian* from Schmidt Ocean Institute (SOI) was used during expedition FKt230629 with R/V *Falkor (too)* and dives S0543-S0560 to explore crustal subseafloor life at the hydrothermal vent site Fava Flow Suburbs at 9°50′N EPR at 2515 m depth in July 2023 (Table 1). Six patches, all of a size of ~ 50 × 50 cm, and located within an area of ~ 20 by 20 meters were studied (Figs. 2, 3). Because the rock was impossible to break into small pieces, we lifted the entire lobate lava shelf. Prior to lifting, a small ~ 2 cm wide hole was drilled with a small ~ 30 cm long chisel to gently widen an existing crack, allowing in situ temperature measurements and sampling of ~ 75 L fluid from the curstal subseafloor using an intake handle (13 mm diameter, WHOI), connected to a miniature magnetic drive gear pump (Suofu; NP060; with 40 W rotor-less brushless direct current motor) operating for 35 min and flexible water tank system (CAN-SB Marine Plastics) mounted on the ROV (Fig. 4). The WHOI intake handle had previously been proven successful to collect subseafloor fluids and microbes[33]. Afterwards, the ~ 2 cm wide hole was enlarged to allow

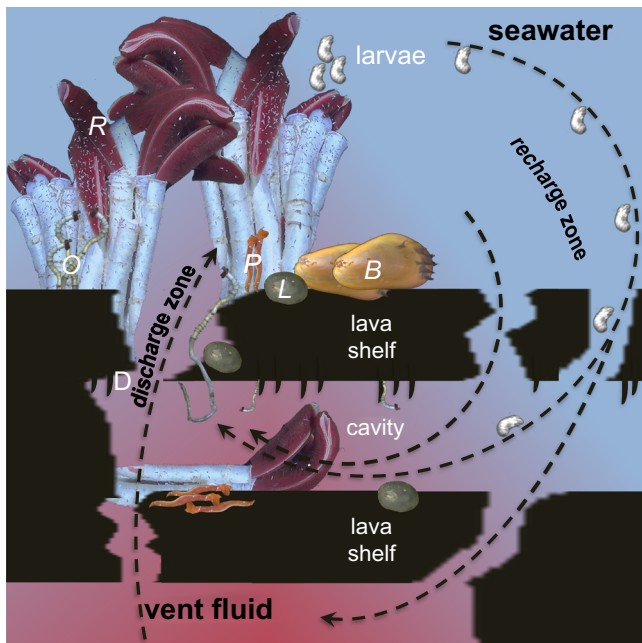

**Fig. 5 | Proposed connectivity model between seafloor surface and crustal subseafloor hydrothermal vents.** Cross-section of lobate lava that is built up by layers of lava shelves with cavities in between. The ceiling of cavities can have lava drips (D). Cracks occur in different localities in the lava shelves. In the recharge zone, cold seawater enters the shallow crustal subseafloor via cracks in the lava shelves and mixes in the subseafloor with upwelling warm hydrothermal vent fluid to be later discharged through cracks in the lava shelves. Seafloor surface vent tubeworms *Riftia pachyptila* (R) and *Oasisia alvinae* (O) release fertilized eggs to develop in the water column into trochophore larvae which get entrained into the crustal subseafloor cavity system in the recharge zone. They settle in the crustal subseafloor, in the cracks, or at the seafloor surface in the discharge zone to grow into adults. Mobile animals, e.g. *Paralvinella* (P) and *Lepetodrilus* (L) either also transit through the circulation system as larvae or migrate in and out of the cracks of the lobate lava. Some animals, e.g. *Bathymodiolus thermophilus* (B) mussels colonize the seafloor surface vents but have not been visually detected in the crustal subseafloor. Scale: lava shelf thickness is ~10 cm, cavities are ~5 to ~15 cm in height, with animals depicted to relative scale (larvae not to scale). The three-dimensional extent of cavities likely reaches several lava shelfs down to lower floors of subsurface.

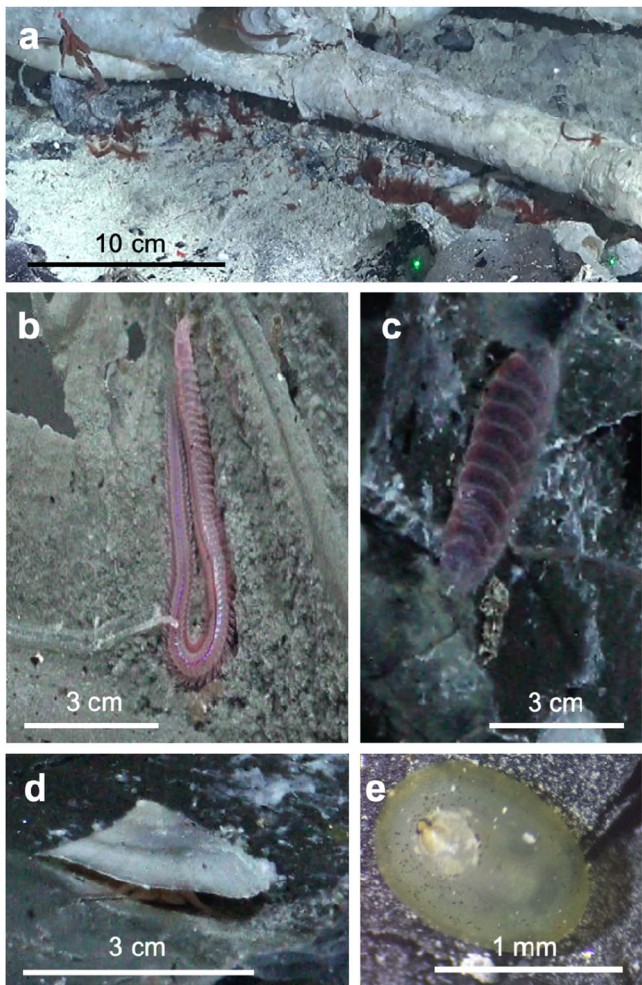

**Fig. 6 | Mobile animals in crustal subseafloor cavities. a** Many *Paralvinella* spp. specimens on top of *Riftia pachyptila* tubes and below at the cavity bottom, green laser points mark 10 cm (dive S0556). **b** *Nereis sandersi* on roof of cavity (dive S0554). **c** *Branchinotogluma* spp. (dive S0554). **d** Semi-sessile *Neomphalus fretterae* (dive S0560). **e** *Lepetodrilus* spp. from roof of cavity (dive S0556). cm = centimeter.

the usage of a suction sampler. Finally, a large 1.5-meter-long pry bar was used to enter the hole diagonally into the cavity for ~80 cm. The lava shelf then was gently lifted and flipped up-side down, opening the view into the subseafloor cavities (Supplementary Movie 1). The whole procedure was filmed, and the bottom and the ceiling of the cavity visually inspected in situ. Short underwater video clips of cavity openings were made with Open Shot Video Editor (OpenShot Studios, LLC, Rockwall, Texas, US). After inspection during ROV dives, parts of the lava shelf were recovered by the ROV arm, placed into a sealed box on the ROV platform, and brought on board of the ship *R/V Falkor(too)*. The lava shelf height was measured with a ruler onboard the ship. Cavity height was estimated in situ based on the distance of ROV lasers.

## Measurements of abiotic water parameters

The ROV *SuBastian* temperature probe PT100 was used to measure in situ temperature of ambient seawater, vent fluid among animal clumps, and vent fluid in the crustal subseafloor cavities (through the ~ 2–3 cm wide hole made with the chisel and in about 20 cm below the seafloor surface). Salinity, pH, oxygen and sulfide concentrations were analyzed from ~ 75 L water samples taken into the flexible water tank system. Onboard the ship the pump was reversed

and water was filled into a glass bottle to analyze pH and salinity with a Multi 340i sensor (WTW, Germany), oxygen concentration together with temperature with the dipping probe DP-PSt3 connected to a Fibox 4 trace oxygen meter (PreSenS, Germany), and $\Sigma H_2S$ (sum of all forms of dissolved sulfide[39],) with the LaMotte sulfide test kit and measured with a HACH Lange DR 1900 photometer calibrated with the UniSense sulfide standard. A non-parametric Kruskal Wallis test was performed to test for any significant differences in temperatures between habitats.

## Macrofaunal analyzes

Underwater videos and framegrabs were analyzed for the presence of vent fauna and identified following Desbruyères et al.[45]. Sizes of tubeworms and other animals were estimated based on ROV laser distance in situ, or measured onboard of the ship directly with a ruler, or taking pictures of animals to compare them with respective scales of an object micrometer with a Canon EOS 550D camera mounted to a BMS Stereo Trino Zoom microscope.

## Reporting summary

Further information on research design is available in the Nature Portfolio Reporting Summary linked to this article.

## Data availability

All data generated or analyzed during this study are included in this published article. All SOI ROV dive videos are open access at youtube (youtube.com/@SchmidtOcean/videos).

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

## Acknowledgements

We thank the captain and crew of R/V *Falkor (too)*, the marine technicians John Fulmer, Julianna Deihl, Tomer Ketter and Tyler Smith onboard and Bob Koster, Edwin Keijzer, Rool Bakker, Jesper van Bennekom and Yetzo de Hoo on shore at NIOZ for their exceptional technical support. This work was funded by Schmidt Ocean Institute and supported by grants from the Rectorate of the University of Vienna and the Austrian Science Fund FWF no. P 3154321 to M.B., and by the Dutch Science Foundation NWO OCENW.M.22.080 to S.G. WHOI Access to the Sea and Investment in Science funds supported S.M.S. WHOI Access to the Sea funds and a NSF graduate fellowship (# 2141064) supported A.F. A WHOI postdoctoral scholarship and an internal WHOI interdisciplinary study award (#27017527) supported C.K. The Slovenian Research Agency (Research Core Funding No. P1-0237) supported T.T. and T.M. We would like to thank Charles R. Fisher for his valuable comments on a previous version of this manuscript.

## Author contributions

M.B. and S.G. designed the in situ experiment and water sampling device, led the ROV dives and field work, and wrote the paper; the performance and innovative support of the ROV pilots (Z.B, R.C., C.H., K.I., A.P.) were crucial for the discovery of the crustal subseafloor cavities and their animals; the cruise participants (A.L.d.O, S.E.-H., A.F., I.V.H., S.H., C.K., I.K., N.K., V.L.L., T.M., A.M., J.M., P.P., I.R.-R., F.S., S.M.S., J.S., T.T., T.R.M.W.) performed lab work, data management, chemical measurements, sample collections, and provided comments and edits on the draft.

## Competing interests

The authors declare no competing interests.
