## [Peer Review File · Nature Communications]

Animal life in the shallow seafloor crust at deep-sea hydrothermal ventsEditorial Note: This manuscript has been previously reviewed at another journal that is not operating a transparent peer review scheme. This document only contains reviewer comments and rebuttal letters for versions considered at *Nature Communications*.

Reviewers' Comments:

Reviewer #1:

Remarks to the Author:

This manuscript describes the discovery of animal life in subsurface habitats, in this case, the lava tubes beneath deep-sea hydrothermal vents. This is a completely novel finding and one that should be of broad interest to the scientific community. I reviewed the previous version of this manuscript, and I find this version to be much more detailed in providing a complete picture of the ecosystem. I found the manuscript very well written and did not find any significant flaws in the analysis or interpretation. I mostly just have some minor comments.

L187-190: These two sentences do not fit at the end of this paragraph. They could conclude the previous paragraph, or could be included in the next section.

Table 1: It would make more sense to have the items in the table in order of dive number and reference in Fig. 1, rather than by their order in the supplementary figure.

Table 1: Why is the minimum H₂S concentration included here? I think a range of values would be more informative.

Figure 2: I understand that there is no consistent scale of the processes described here, but some sense of the distances involved would be helpful. In particular, what is the range of distances for the recharge zone? I believe that it is on the scale of 100s of meters to kilometers. Similarly, the vent fluid needs to be fairly deep in the subsurface to interact with the magma and gain heat energy. The way it is presented here, it looks like all of the processes are on the cm to m scale, like the organisms pictured.

Supplementary Text / Experimental Design: I don't see any other reference to these experiments in this manuscript, and this section should be removed.

Reply to Reviewer1:

Reviewer #1 (Remarks to the Author):

This manuscript describes the discovery of animal life in subsurface habitats, in this case, the lava tubes beneath deep-sea hydrothermal vents. This is a completely novel finding and one that should be of broad interest to the scientific community. I reviewed the previous version of this manuscript, and I find this version to be much more detailed in providing a complete picture of the ecosystem. I found the manuscript very well written and did not find any significant flaws in the analysis or interpretation. I mostly just have some minor comments.

R: we thank the reviewer for the positive evaluation.

L187-190: These two sentences do not fit at the end of this paragraph. They could conclude the previous paragraph, or could be included in the next section.

R: We agree and moved the sentence to the next paragraph (inserted after third sentence).

Table 1: It would make more sense to have the items in the table in order of dive number and reference in Fig. 1, rather than by their order in the supplementary figure.

R: We agree and reordered the table according to dive numbers and Figure 1.

We note that in the Authors checklist the editors suggested to move all figures from the supplements to the main text. We agree and thus changed throughout the manuscript the figure numbers. We note that the order of pictures (a,b,c,d,...) in supplementary figure 2 is therefore slightly changed so it follows the same order as in Figure 1 and Table 1 (recovery dive 1 to 6).

Table 1: Why is the minimum H₂S concentration included here? I think a range of values would be more informative.

R: as stated in the methods, we measured sulfide once per subsurface (collection of water via a pump) and thus cannot provide a range of values per dive. A minimum value is typically provided, to account for any potential autooxidation with oxygen during sample processing.

Figure 2: I understand that there is no consistent scale of the processes described here, but some sense of the distances involved would be helpful. In particular, what is the range of distances for the recharge zone? I believe that it is on the scale of 100s of meters to kilometers. Similarly, the vent fluid needs to be fairly deep in the subsurface to interact with the magma and gain heat energy. The way it is presented here, it looks like all of the processes are on the cm to m scale, like the organisms pictured.

R: the scale we observed in situ is on several cm to meter level, and the reviewer is right that the processes are on much larger scale. To be make this clearer, we added some text (underlined below) in the figure legend on scale (the overall scale itself is discussed in more detail the manuscript text).

"Scale: lava shelf thickness is ~10 cm, cavities are ~5 to ~15 cm in height, with animals depicted to relative scale (larvae not to scale). The three-dimensional extent of cavities likely reaches several lava shelves down to lower floors of subsurface.

Supplementary Text / Experimental Design: I dont see any other reference to these experiments in this manuscript, and this section should be removed.

R: we agree with the reviewer and have removed this part.